# Efficacy of Self-Management on Glucose Control in Type 2 Diabetes Mellitus Patients Treated with Insulin

**DOI:** 10.3390/healthcare10102080

**Published:** 2022-10-19

**Authors:** Hsiu-Chu Lin, Chin-Wei Tseng, Ping-Jung Hsieh, Hsiu-Ling Liang, Shu-Ping Sue, Chun-Yang Huang, Mei-Yueh Lee

**Affiliations:** 1Division of Endocrinology and Metabolism, Department of Internal Medicine, Kaohsiung Medical University Hospital, Kaohsiung 807, Taiwan; 2Department of Nursing, Kaohsiung Medical University Hospital, Kaohsiung 807, Taiwan; 3Graduate Institute of Human Resource and Knowledge Management, National Kaohsiung Normal University, Kaohsiung 802, Taiwan; 4Faculty of Medicine, College of Medicine, Kaohsiung Medical University, Kaohsiung 807, Taiwan

**Keywords:** diabetes mellitus, self-management, insulin treatment

## Abstract

Diabetes mellitus (DM) is a chronic metabolic disease. If blood glucose is poorly controlled, it will cause a variety of chronic complications. Therefore, the issue of healthcare in diabetic patients is a problem that cannot be ignored. In this study, we aim to investigate the correlation between sociodemographic characteristics, self-management, and glycated hemoglobin (HbA1c) values in patients with type 2 diabetes treated with insulin. A total of 300 type 2 diabetic patients treated with insulin were enrolled. Type 2 diabetic patients treated with insulin had a significant negative correlation of HbA1c value to self-management total score. The lower the HbA1c value, the better the self-management of type 2 diabetic patients treated with insulin is. It is recommended that scale assessment tools be used to identify problems, improve the self-management ability of type 2 diabetic patients, and problem solve in patients in order to facilitate the effectiveness of blood glucose control of type 2 diabetic patients.

## 1. Introduction

According to statistics for 2019 from the Ministry of Health and Welfare of the Executive Yuan in Taiwan, diabetes was the fifth out of ten leading causes of death, and the number of deaths was as high as 9996 [1]. The healthcare of diabetic patients is a health issue that cannot be ignored in the current society.

Diabetes mellitus (DM) is a chronic metabolic disease. If blood glucose is not well controlled, it will cause a variety of chronic complications, including retinopathy, nephropathy, neuropathy, and cardiovascular issues such as myocardial infarction, stroke, and peripheral arterial disease, which can directly affect personal safety and quality of life. Therefore, preventing and delaying the occurrence of these micro- and macrovascular complications is actually an important indicator of the effectiveness of diabetes care [2]. A study pointed out that in diabetic patients with micro- or macrovascular complications, the medical expenses are four times higher than those of diabetic patients without vascular complications in Taiwan [3]. Therefore, the issues of personal and family care, social support, and medical expenses in diabetes control are quite important.

The β-cell dysfunction is present at the diagnosis of type 2 DM and progressively worsens with disease duration. β-cell dysfunction is associated with worsening glycemic control and treatment failure. When combined use of multiple oral hypoglycemic drugs cannot achieve the optimal therapeutic effect, insulin therapy should be considered [4]. In addition, most people with type 2 diabetes ultimately need insulin therapy. When blood glucose control is poor, the need for insulin to control blood glucose should be explained objectively to the patient while avoiding treating it as a threat or as a punishment. While emphasizing the disease progression of diabetes mellitus, the effectiveness, and importance of insulin treatment for maintaining blood glucose stability is superior to other blood glucose-lowering medications [5]; therefore, insulin therapy is an effective way to control blood glucose in type 2 diabetic patients.

However, some studies have shown type 2 diabetic patients under insulin treatment experience internal and external struggles, and resist dealing with negative emotions. In such cases, the health professional should advise patients of counseling opportunities and implement a program to provide knowledge of proper care [6]. Some studies also mention in the process of long-term diabetes control, in addition to assisting blood glucose control through medication and lifestyle modification, it is necessary to pay attention to the patient’s participation in the management of chronic diseases and the degree of self-management [7]. Studies have also shown that the basic elements of the chronic care model are self-management which can affect the effectiveness of blood glucose control in type 2 diabetic patients [8]. The most important goal of self-management for a diabetic patient is to control blood glucose and glycated hemoglobin (HbA1c), reduce the occurrence of complications, prevent vascular diseases, and have a good quality of life [9,10].

When type 2 diabetic patients need the intervention of insulin for better blood glucose control, their willingness to self-manage may affect the change of HbA1c. Although there are current studies on self-management and blood glucose control in diabetic patients [9], there are only limited studies on the association of self-management and HbA1c in type 2 diabetic patients on insulin therapy. In our study, we aim to investigate the efficacy of self-management total score on the HbaA1c value in type 2 diabetic patients on insulin therapy. We expect self-management total score can be used as an effective evaluation tool for medical personnel in clinical practice and health education to provide the most appropriate care and services and to improve the effectiveness of blood glucose control in type 2 diabetic patients on insulin treatment.

## 2. Materials and Methods

### 2.1. Research Design and Objectives

This study adopted a cross-sectional collection and a deliberate sampling design. The data were collected in the Endocrinology and Metabolism outpatient department of a medical center in Southern Taiwan. Patients with type 2 diabetes who had been treated with insulin, who were over 20 years old, had a clear consciousness; and who were able to communicate in Mandarin or Taiwanese were enrolled. Patients using insulin therapy who were in acute infections, had cognitive problems, declined to participate in this study, or had recently been hospitalized due to hyperglycemia were excluded. A total of 300 patients agreed to join the study after a briefing, filled in the questionnaire consent form, provided basic information, and answered the diabetes self-management scale included (Figure 1). F-tests were used for sample size calculation. The actual power revealed 80.31%.

### 2.2. Laboratory Analysis

Fresh whole blood samples collected with EDTA anticoagulant were obtained for glucose measurement by a licensed medical technologist at our laboratory. HbA1c was measured on the blood-sampling days by enzyme immunoassay (Determiner, Kyowa Medex Co., Ltd., Tokyo, Japan).

### 2.3. Research Tools

In this study, data were collected via an anonymous structured questionnaire. The content of the questionnaire included the sociodemographic characteristics of the patient and the diabetes self-management scale. The scale was used with the permission of the author. The questionnaire is mainly based on the patient’s statements, and the glycosylated hemoglobin value is obtained from the latest electronic medical record.

(1)The sociodemographic characteristics of the patient

The sociodemographic characteristics were collected for the patient’s gender, age, duration of diabetes mellitus, education level, marital status, occupation, height, weight, body mass index (BMI), current insulin treatment, and health status.

(2)Diabetes Self-Management Instrument Short Form (DSMI-20)

This scale adopts the Diabetes Self-Management Scale [11]. The reason for the adoption is that the number of questions in the scale is appropriate, is easy to understand, can be completed in a short time, the content is less privacy-related, and it is easy for patients to fill in the information. The degree is high and easy to perform in the outpatient department.

There are 20 items on the scale, including four factors: “Medical Partnership”, “Daily Care”, “Glucose Goal Achievement”, and “Problem Solving”. The four-point Likert scale was used to evaluate each item. The respondent’s own feelings about diabetes self-management and how to deal with it should be answered, from 1 point to “never” for not every day or not every time to 4 points for every day or almost every time. “Always” is calculated based on the average score of a single item. All items are positive vocabulary questions, with a total score of 80 points. The higher the score, the better the individual’s self-management of diabetes is.

In terms of the reliability and validity of the scale, the internal consistency (Cronbach’s α) of DSMI-20 is 0.925, and the internal consistency of the four factors is between 0.838 and 0.892. The retest reliability of DSMI-20 implemented at 2-week intervals is r = 0.790 (*p* < 0.001), indicating that the scale has good reliability and validity [11].

### 2.4. Data Collection

This study was reviewed by the Institutional Review Board of Kaohsiung Medical University Hospital with the IRB number KMUHIRB-E(Ⅱ)-20190373. After the approval, the acceptance of the study population began. The scope of the study includes people eligible from 10 March 2020, to 21 April 2020. Those who met the qualifications were enrolled in this study. The investigator explained the purpose of the study and asked for the consent and signature of the study population; the investigator collected the basic information and surveys for the diabetes self-management scale. When collecting data, attention was given to the surrounding environment to maintain patient privacy and provide sufficient time for the patient to read and complete the questionnaire. For patients who were illiterate or had difficulty reading, the investigator explained the content of the questionnaire at an appropriate sound volume and speed according to the content of the study populations’ answers and helped them fill in the questionnaire.

### 2.5. Data Processing and Analysis

The questionnaire data are presented as the mean and standard deviation for continuous variables and the proportions for categorical variables. The difference between gender, marital status, and occupation were compared by two sample tests, education level was used as a one-way analysis of variance (ANOVA), and the correlative in continuous variables used the Pearson correlation coefficient to analyze the results.

Age and significant variables in univariate analysis were selected for the multiple regression analysis. The study used SPSS 24.0 statistical software (IBM, Armonk, NY, USA) for data archiving and statistical analysis. A *p* value less than 0.05 is statistically significant.

## 3. Results

A total of 300 cases were enrolled in this study, including 151 males (50.3%) and 149 females (49.7%), with a mean age of 62.1 ± 11.1 years. The mean duration of diabetes mellitus was 16.7 ± 9.2 years. The education level of the study population was mostly high school/high vocational (34.7%), married (76.3%), and unemployed or retired (62.3%). The mean HbA1c value was 8.0% ± 1.62%. In terms of self-perceived health status (0–10 points), the mean score was 6.4 ± 1.7. The higher the score was, the more satisfactory the self-perceived health status is (Table 1).

There were no significant differences among the study population in gender, education level, and marital status to HbA1c value. In terms of occupation, the HbA1c value of 7.79 ± 1.53% for unemployed (including retirees) was better than the HbA1c value of 8.03 ± 1.72% for blue-collar workers and white-collar workers of 8.58 ± 2.29%, indicating that unemployed people have better glucose control than those employed (Table 2).

The mean age of 62.14 ± 11.13 and the HbA1c value are significantly negatively correlated, showing that the older the age, the lower the HbA1c value. The self-perceived health status score of 6.43 ± 1.71 reached a significant negative correlation difference with the HbA1c value, which means that the higher the score of self-perceived health status is, the lower the HbA1c value. However, there was no correlation between the HbA1c value with the duration of diabetes mellitus and body mass index (Table 3).

Through the different categories of self-management total score, the HbA1c value has a significant negative correlation with medical partnership (3.11 ± 0.78, *p* < 0.001), daily life care (3.09 ± 0.75, *p* < 0.001), blood glucose goal achievement (3.01 ± 0.85, *p* < 0.001), and problem solving (2.98 ± 0.77, *p* < 0.001). The self-management total score (60.96 ± 14.06, *p* < 0.001) is significantly negatively correlated with the HbA1c value, indicating that the better the self-management of the study population, the lower the HbA1c value (Table 4).

In multivariate analysis, after adjusting to age, self-perceived health status, occupation, and self-management total score, the self-management total score [β = −0.02 (−0.04, −0.01), *p* value =< 0.001], age [β = −0.03 (−0.05, −0.01), *p* value =< 0.001], and self-perceived health status [β = −0.15 (−0.26, −0.04), *p* value =< 0.001] persistently show a significant correlation with HbA1c value. However, the occupation did not show a significant correlation with the HbA1c value (Table 5).

## 4. Discussion

Our study shows that the total score of self-management of type 2 diabetes mellitus under insulin treatment is significantly negatively correlated to the HbA1c value. A higher self-management total score has a better HbA1c value, which is the same as the previous results [12,13,14]. Many researchers have different definitions of self-management in their research, but most believe that self-management includes behavior-oriented healthcare [15,16,17]. Self-management healthcare activities include medical management, symptom treatment, psychological problem-solving; openness to social assistance and support, and lifestyle modification when necessary [15]. Self-management behavior of diabetic patients must factor in the duration of diabetes, and treatment options including nutritional intervention, increased physical activity, medical management, and prevention and treatment of comorbidities. Patients should formulate personal strategies to solve social and psychological problems and promote health and behavior changes [16]. Previous studies have shown that self-management is varied in different healthcare strategies, including diet, exercise, medication, symptom treatment, skills in handling emotions and communications, and other psychological and social issues [15,16,17]. Chronic disease patients can achieve effective self-management through good communication in daily life, correct symptom management, and partnership with caregivers [17]. Self-management and medical partnership are positively related; therefore, the medical partnership is also an important part of self-management [18]. In our study, the self-management total score included the four major categories of medical partnership, daily life care, blood glucose goal achievement, and problem-solving.

In a study on chronic disease self-management, it was pointed out that chronic disease patients must be able to monitor their own symptoms and know how to make use of community resources. Through communication with physicians, symptom management is part of the process of self-management [19]. In our study, the self-perceived health status score was negatively correlated with the HbA1c value. The higher the self-perceived health status score was, the lower the HbA1c. 

HbA1c is an indicator of glucose control. After confirming self-management, the effectiveness of HbA1c can assist healthcare workers to strengthen interventional measures for patient self-management [20]. A cross-sectional study on patients with type 2 diabetes mellitus found that if patients can exercise regularly and have better diet control behaviors, they can have lower HbA1c levels and better blood glucose control [20]. A longitudinal survey of people with type 2 diabetes found that those with better self-management have lower HbA1c levels and better glucose levels [12]. The results of the study showed that the HbA1c value of type 2 diabetic patients improved significantly after regular self-management assessment [13]. A cross-sectional survey of elderly people over 65 years of age with type 2 diabetes shows that age and diabetes self-management total scores are significantly negatively correlated with HbA1c [14]. The better the self-management of the elderly is, the better their blood glucose control. However, in our study, the self-management total score was not significantly related to the HbA1c value in the elderly subgroup. 

In our study, the mean HbA1c value was 8.0 ± 1.7%. This is higher than the results of the other study where the average HbA1c value of patients with type 2 diabetes was at 7.6% [21]. The causal inference is that type 2 diabetes is a chronic disease, and long-term diet control, regular exercise, and medical treatment are needed to maintain a healthy lifestyle and achieve good blood glucose control although this might be impossible for some patients. 

In terms of the comparative analysis of socio-demographic characteristics and the HbA1c value of the study population, gender, marital status, and duration of diabetes mellitus have no effect on HbA1c value, which are consistent with our study [14]. The statistical analysis showed a significantly negative correlation of age to Hba1c value; the older the age, the better the blood glucose control is, which is consistent with the previous results [14,22]. Occupation and HbA1c values reach a statistically significant correlation, representing a lower HbA1c value and better blood glucose control in the unemployed or retired population, and this result is consistent with a previous study [23]; however, it did not reach a significant correlation on the efficacy of the self-management total score to HbA1c value. Finally, there is no statistically significant correlation between the comparison of body mass index and HbA1c value, which means that the body weight of the study population is not significantly related to blood glucose control, which is inconsistent with the previous results. Some studies have shown that body mass index is related to blood glucose control. The higher the body mass index, the more difficult it is to control blood glucose. This may be the case for the specific ethnic group of patients with type 2 diabetes mellitus under insulin treatment, leading to a different result [24].

The strength of our study is the actual power of the sample size which is large enough to reach a significant correlation of the efficacy of the self-management total score on the HbA1c value in type 2 diabetes mellitus patients treated with insulin. The limitation of our study is that we did not evaluate the adherence and compliance to insulin use. Second, this is a cross-sectional study. A long-term follow up by repeating the questionnaire in the same study population after a period might be necessary to verify whether the self-management total score of the study population will be higher after reevaluation. Lastly, socioeconomic status might be an important factor to affect the self-management total score. However, because of confidentiality concerns about income status by the study population, we were not able to provide detailed information about socioeconomic status, and we replaced socioeconomic status by unemployed or retired and white or blue-collar category instead.

## 5. Conclusions

It is recommended that relevant assessment tools such as self-management scales be used in daily clinical practice to detect patients’ problems in daily healthcare early. Patients should also be aware of hyperglycemia and hypoglycemia and have insights on their own blood glucose monitoring. With these self-management scales, healthcare professionals could design individualized blood glucose control strategies for appropriate patients.

## Figures and Tables

**Figure 1 healthcare-10-02080-f001:**
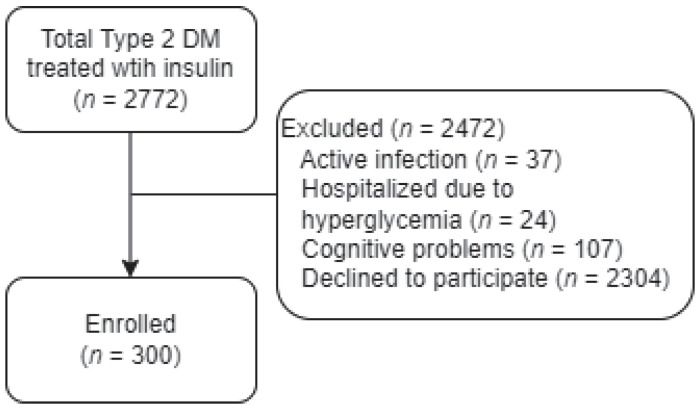
The flow diagram of the patients participating in this study. DM, diabetes mellitus.

**Table 1 healthcare-10-02080-t001:** The demographic characteristics of the study population.

Variation	Number (*n* = 300)	%	Mean ± SD
**Gender**			
Male	151	50.3%	
Female	149	49.7%	
**Age** Mean			62.1 ± 11.13
20–30 years old	1	0.3%	
31–40 years old	14	4.7%	
41–50 years old	28	9.3%	
51–60 years old	75	25.0%	
61–70 years old	108	36.0%	
>70 years old	74	24.7%	
**Duration of diabetes mellitus** Mean			16.7 ± 9.18
0–10 years	79	26.3%	
11–20 years	121	40.3%	
21–30 years	81	27.0%	
>30 years	19	6.4%	
**Education level**			
Below junior high school	117	39.0%	
High school/Vocational high school	104	34.7%	
Bachelor/Above Bachelor’s degree	79	26.3%	
**Marital status**			
Married	229	76.3%	
Single/Widowed/Divorce	71	23.7%	
**Occupation**			
Unemployed/Retired	187	62.3%	
Blue-collar workers	82	27.3%	
White-collar workers	31	10.3%	
**Height (cm)**			162.4 ± 8.82
**Weight (kg)**			71.4 ± 13.01
**Body mass index (BMI)**			26.9 ± 3.94
**HbA1c (%)**			8.0 ± 1.62
**Self-perceived health status** **(0–10 points)**			6.4 ± 1.71

HbA1c, glycated hemoglobin.

**Table 2 healthcare-10-02080-t002:** The analysis of the demographic characteristics and the HbA1c value.

Variation	*n* = 300	HbA1c	*p*
		Mean ± SD	
**Gender**			0.588
Male	151	7.90 ± 1.63	
Female	149	8.01 ± 1.75	
**Education level**			0.940
Below junior high school	117	7.91 ± 1.56	
High school/Vocational high school	104	7.96 ± 1.77	
Bachelor/Above Bachelor’s degree	79	8.00 ± 1.78	
**Marital status**			0.364
Married	229	7.90 ± 1.60	
Single/Widowed/Divorce	71	8.11 ± 1.95	
**Occupation**			0.038 *
Unemployed/ Retired	186	7.79 ± 1.53	
Blue-collar workers	82	8.03 ± 1.72	
White-collar workers	31	8.58 ± 2.29	

Hba1c, glycated hemoglobin; * *p* < 0.05.

**Table 3 healthcare-10-02080-t003:** The analysis of continuous variables and the HbA1c value.

Variation			HbA1c		
			Mean ± SD	Pearson Correlation	*p* (Two-Tailed)
Age	62.14 ± 11.13	−0.252 ***	0.000 ***
Duration of diabetes mellitus	16.68 ± 9.18	0.013	0.825
Body mass index	26.93 ± 3.94	0.109	0.060
Self-perceived health status score (0–10)	6.43 ± 1.71	−0.221 ***	0.000 ***

HbA1c, glycated hemoglobin. *** *p* < 0.001.

**Table 4 healthcare-10-02080-t004:** The analysis of the self-management total score and the HbA1c value.

Variation			HbA1c		
			Mean ± SD	Pearson Correlation	*p* (Two-Tailed)
Medical partnership	3.11 ± 0.78	−0.223 ***	0.000 ***
Daily life care	3.09 ± 0.75	−0.238 ***	0.000 ***
Blood glucose goal achievement	3.01 ± 0.85	−0.245 ***	0.000 ***
Problem-solving	2.98 ± 0.77	−0.234 ***	0.000 ***
Total score	60.96 ± 14.06	−0.264 ***	0.000 ***

HbA1c, glycated hemoglobin; *** *p* < 0.001.

**Table 5 healthcare-10-02080-t005:** Multivariate analysis of HbA1c value after adjusting for age, self-perceived health status, occupation, and self-management total score.

Variation	Coefficient (β)	95% CI	*p* (Two-Tailed)
**Intercept**	12.51	(10.98, 14.03)	0.000 ***
**Age**	−0.03	(−0.05, −0.01)	0.000 ***
**Self-perceived health status** **(0–10 points)**	−0.15	(−0.26, −0.04)	0.000 ***
**Occupation** Unemployed/Retired (Reference)			
Blue-collar workers	−0.11	(−0.57, 0.35)	0.631
White-collar workers	0.23	(−0.41, 0.87)	0.474
**Self-management total score**	−0.02	(−0.04, −0.01)	0.000 ***

*** *p* < 0.001.

## Data Availability

Not applicable.

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
