# Peer review of "Efficacy of Self-Management on Glucose Control in Type 2 Diabetes Mellitus Patients Treated with Insulin"

_healthcare, 2022, doi:10.3390/healthcare10102080_

Round 1

Reviewer 1 Report

Title:

The association of insulin treatment for self-management and glycemic control is not identified. This study deals with self-management and glycemic control in patients with type 2 diabetes mellitus with insulin treatment. The title is confusing with the rest of the work.

Abstract:

The evaluation of emotional discomfort is mentioned as an objective, but it is not mentioned in the methodology and results.

Introduction

In paragraph 4 of the introduction, the information is confusing. It is described that patients prefer oral treatment instead of injectable. But this work involves patients who have insulin treatment. It should focus on situations experienced by patients with this treatment that imply poor glycemic control.

Do not mix self-management and self-care, they are two different concepts. Focus in the introduction only on self-management.

It is necessary to describe the experiences of patients who have insulin treatment that may affect self-management and glycemic control, because they indicate that the use of insulin can achieve glycemic control.

Verify what the objective of the investigation will be. Here it is mentioned that they want to identify the factors that affect glycemic control and self-management, but the results only focus on glycemic control.

Methods

It is not mentioned how the HbA1c was evaluated, what technique was used to process the blood sample, who, where or at what time the test was performed. Only the application of the questionnaires is mentioned.

In the statistical analysis, one-way ANOVA and Pearson's correlation are mentioned as inferential analyses, but comparisons of two independent groups are observed. Missing analyzes should be added.

Discussion.

They do not discuss the results.

They do not point out the weakness of the results because adherence to treatment (insulin) was not evaluated.

Author Response

The association of insulin treatment for self-management and glycemic control is not identified. This study deals with self-management and glycemic control in patients with type 2 diabetes mellitus with insulin treatment. The title is confusing with the rest of the work.

Response: Thank you for your comment. We now revised our title.

Abstract:

The evaluation of emotional discomfort is mentioned as an objective, but it is not mentioned in the methodology and results.

Response: We deleted the “emotional discomfort” in our abstract and focus on the items of the questionnaire.

Introduction

In paragraph 4 of the introduction, the information is confusing. It is described that patients prefer oral treatment instead of injectables. But this work involves patients who have insulin treatment. It should focus on situations experienced by patients with this treatment that imply poor glycemic control.

Response: The description of the patients with oral treatment was already deleted.

Do not mix self-management and self-care, they are two different concepts. Focus in the introduction only on self-management.

Response: Yes, we uniform the term “self-care” and “self-management” to self-management now.

It is necessary to describe the experiences of patients who have insulin treatment that may affect self-management and glycemic control, because they indicate that the use of insulin can achieve glycemic control.

Response: It’s now well described in lines 59-68.

Verify what the objective of the investigation will be. Here it is mentioned that they want to identify the factors that affect glycemic control and self-management, but the results only focus on glycemic control.

Response: Our study objective was now verified in lines 76-81.

Methods

It is not mentioned how the HbA1c was evaluated, what technique was used to process the blood sample, who, where or at what time the test was performed. Only the application of the questionnaires is mentioned.

Response: The evaluation of the HbA1c was now added in the method section as 2.2 Laboratory analyses (lines 101-104).

In the statistical analysis, one-way ANOVA and Pearson's correlation are mentioned as inferential analyses, but comparisons of two independent groups are observed. Missing analyzes should be added.

Response: The description of the statistical analysis was revised. (Lines 150-157)

Discussion.

They do not discuss the results.

Response: We do the discussion of our results on every paragraph now.

They do not point out the weakness of the results because adherence to treatment (insulin) was not evaluated.

Response: We added the strength and limitations of our study in the last paragraph of the discussion.

Reviewer 2 Report

This study examines the barriers to self-glycemic control. I have read this carefully and with interest.

1. Would you provide the English version of the questionnaire used in this study as supplementary figure?

2. I thought it would be better if there was a reference to the results of the sub-analysis restricted to the elderly (Table 4).

3. "HbA1c" and "HbA1C" co-existed.

I did not realize any problem with this otherwise.

Author Response

This study examines the barriers to self-glycemic control. I have read this carefully and with interest.

  1. Would you provide the English version of the questionnaire used in this study as supplementary figure?

Response: Yes, we now provided it as a supplementary figure now.

  1. I thought it would be better if there was a reference to the results of the sub-analysis restricted to the elderly (Table 4).

Response: Thank you for your comment. We did a subgroup analysis for the elderly, but the results didn’t come out to be significant. We attached the analyses for your reference below.

Table. The subgroup analyses of the elderly on HbA1c to self-management score

HbA1c

Variation

Mean±SD

Pearson correlation

P(two-tailed)

Medical partnership

3.15±0.75

-0.075

0.3837

Daily life care

3.14±0.77

-0.107

0.2171

Blood glucose goal achievement

3.10±0.88

-0.104

0.2263

Problem-solving

3.02±0.82

-0.083

0.3363

Total score

62.08±14.32

-0.102

0.2354

Hba1c, glycated hemoglobin.

*P< .05、**P< .01、***P< .001

  1. "HbA1c" and "HbA1C" co-existed.

Response: We now uniformed the “HbA1c “ in the whole manuscript.

I did not realize any problem with this otherwise.

Reviewer 3 Report

The manuscript entitled “Association of insulin treatment to self-management and glucose control in type 2 Diabetes Mellitus patient” describes the assessment of the self-management glycemic control in 300 patients with type 2 diabetes (T2D). Although the authors highlight the use of insulin in glycemic control, the study is not focused on the action of this hormone in the self-management of T2D because all participants only received insulin as therapy without comparing it to other hypoglycemic or anti-hyperglycemic drugs. For this reason, I suggest the authors change the rationale behind the study and make a deep manuscript restructuration. Please find below the point-by-point review report that the authors should reply to before the manuscript can be considered for publication.

Also, the manuscript contains numerous mistakes in English grammar, tone, and punctuation that a native English speaker must fix.

ABSTRACT

-Please reorganize the abstract describing the results and narrow the main conclusion to the findings in more detail.

INTRODUCTION

-In my viewpoint, the study is not focused on the action of insulin in the self-management of glycemic control because all participants only received insulin as therapy without comparing it to other hypoglycemic or anti-hyperglycemic drugs. For this reason, I suggest the authors change the rationale behind the study. Also, shorten the introduction because it is too long and repetitive.

MATERIAL AND METHODS

-Following the last question, please add insulin treatment to the inclusion criteria.

-Please provide more detail regarding the statistical analyses you performed.

-Did you measure the consistency between participants' responses in the 20-query questionnaire and what they did regarding the self-management of glycemic control?

-Provide sample size calculation showing that n = 300 is enough to make solid conclusions.

-I suggest you add the exact questions of the questionnaires as a supplementary file.

RESULTS

-There is no information regarding the type of job or occupation. Some jobs allow more free time for patients to take care of their therapy.

-Please add information regarding socioeconomic status because high-income patients tend to show better self-management of glycemic control than low-income patients.

-Please add a figure showing the selection process of participants according to the Consolidated Standards of Reporting Trials (CONSORT) guidelines.

-I strongly suggest the authors perform multivariate analyses to assess the possible contribution of variables such as education level, occupation, or socioeconomic status to effective self-management of glycemic control.

DISCUSSION

-Please refocus the discussion considering all questions above; overall, the study is not focused on the action of insulin in the self-management of glycemic control because all participants only received insulin as therapy.

CONCLUSIONS

-Please shorten the study’s conclusion by focusing on the main findings.

Author Response

The manuscript entitled “Association of insulin treatment to self-management and glucose control in type 2 Diabetes Mellitus patient” describes the assessment of the self-management glycemic control in 300 patients with type 2 diabetes (T2D). Although the authors highlight the use of insulin in glycemic control, the study is not focused on the action of this hormone in the self-management of T2D because all participants only received insulin as therapy without comparing it to other hypoglycemic or anti-hyperglycemic drugs. For this reason, I suggest the authors change the rationale behind the study and make a deep manuscript restructuration. Please find below the point-by-point review report that the authors should reply to before the manuscript can be considered for publication.

Also, the manuscript contains numerous mistakes in English grammar, tone, and punctuation that a native English speaker must fix.

ABSTRACT

-Please reorganize the abstract describing the results and narrow the main conclusion to the findings in more detail.

 Response: We now revised our abstract to a simple but detailed description of our findings.

INTRODUCTION

-In my viewpoint, the study is not focused on the action of insulin in the self-management of glycemic control because all participants only received insulin as therapy without comparing it to other hypoglycemic or anti-hyperglycemic drugs. For this reason, I suggest the authors change the rationale behind the study. Also, shorten the introduction because it is too long and repetitive.

Response: We now do a revision of our introduction which focuses on the self-management of insulin-treated type 2 diabetes patients. The introduction now is more simplified and not repetitive.

MATERIAL AND METHODS

-Following the last question, please add insulin treatment to the inclusion criteria.

Response: Yes, the insulin treatment is added already.

-Please provide more detail regarding the statistical analyses you performed.

Response: We do a revision for a detailed statistical analysis in method section ((Lines 150-159)

-Did you measure the consistency between participants' responses in the 20-query questionnaire and what they did regarding the self-management of glycemic control?

Response: The participants ‘ responses were well described now in the method section.

-Provide sample size calculation showing that n = 300 is enough to make solid conclusions.

Response: F-tests were used for sample size calculation. The actual power revealed 0.80, with a total sample size of 73 will be significant for our study.

-I suggest you add the exact questions of the questionnaires as a supplementary file.

Response: We now provided the English version of the questionnaires in a supplementary file.

RESULTS

-There is no information regarding the type of job or occupation. Some jobs allow more free time for patients to take care of their therapy.

Response: We now recategorized the occupation to unemployed/retired, white and blue collar, and also to represent the income status.

-Please add information regarding socioeconomic status because high-income patients tend to show better self-management of glycemic control than low-income patients.

Response: We now recategorized the occupation to unemployed/retired, white, and blue collar. Due to concerns of the study population about the confidentiality of the information about income status, we replaced the income status with the white and blue-collar categories in occupation. And this was listed in our limitation of the study.

-Please add a figure showing the selection process of participants according to the

Consolidated Standards of Reporting Trials (CONSORT) guidelines.

Response: We added it as figure 1 now.

-I strongly suggest the authors perform multivariate analyses to assess the possible contribution of variables such as education level, occupation, or socioeconomic status to effective self-management of glycemic control.

Response: The multivariate analysis was performed and added as Table 5.

DISCUSSION

-Please refocus the discussion considering all questions above; overall, the study is not focused on the action of insulin in the self-management of glycemic control because all participants only received insulin as therapy.

Response: We now do a major revision to our discussion section mainly focusing on insulin- treated type 2 diabetes mellitus patients, please reconsider.

CONCLUSIONS

-Please shorten the study’s conclusion by focusing on the main findings.

Response: We now shorten our conclusion and focus on the main findings.

Round 2

Reviewer 1 Report

- In the summary delete "Evaluation was based on questionnaires and 20 questions diabetes self-management scale" includes the name of the diabetes self-management questionnaire.

-In Table 2, you must include what they mean or to which statistical analysis they correspond (F/t).

-Include in data analysis that you used student's t for comparison of two groups.

-Delete from the Data processing "F-tests were used for sample size calculation. The actual power revealed 0.80, with a total sample size of 73 will be significant for our study." Instead, perform a recalculation of the power size with the sample you obtained, and place the results at the end of the first methods paragraph.

Author Response

 In the summary delete "Evaluation was based on questionnaires and 20 questions diabetes self-management scale" includes the name of the diabetes self-management questionnaire.

Response: The statement "Evaluation was based on questionnaires and 20 questions diabetes self-management scale" was already deleted from the abstract.

-In Table 2, you must include what they mean or to which statistical analysis they correspond (F/t).

Response: We consulted other expert statistician, as her suggestion, we revised the presentation of the Table 2.

-Include in data analysis that you used student's t for comparison of two groups.

Response: We now explained the data analysis for student T test and ANOVA at the data processing and analysis of method section.

-Delete from the Data processing "F-tests were used for sample size calculation. The actual power revealed 0.80, with a total sample size of 73 will be significant for our study." Instead, perform a recalculation of the power size with the sample you obtained, and place the results at the end of the first methods paragraph.

Response: The sample power size result now was added at the end of the first methods paragraph. The statement “F-tests were used for sample size calculation. The actual power revealed 0.80, with a total sample size of 73 will be significant for our study” was now deleted from statistic analysis.

Reviewer 3 Report

The authors considered all questions and concerns for the previous review report. The manuscript is now improved.

Author Response

The authors considered all questions and concerns for the previous review report. The manuscript is now improved.

Response: Thank you for your consideration and response.